child depression; child depression screening tool; Sub-Saharan Africa

**Corresponding author:**
Sharain Suliman;
Email: sharain@sun.ac.za

# Validation of the Child Depression Screening Tool in three African settings: Rwanda, Senegal and South Africa

Sharain Suliman[1] , Jenny Bloom[1], Naeem Dalal[2,3] , Eric Remera[4],
Raissa Muvunyi[4], Mohammed Abdulaziz[5], Adelard Kakunze[3],
Ismahan Soukeyna Diop[6], Djena Fafa Cisse[7], Ndeye Awa Dieye[7], Britt McKinnon[8],
Mohamadou Sall[9], Agnes Binagwaho[10,11] and Soraya Seedat[1]

[1]MRC Genomics of Brain Disorders Unit and Department of Psychiatry, Stellenbosch University, Cape Town, South Africa; [2]Zambia National Public Health Institute, University of Zambia School of Medicine, Lusaka, Zambia; [3]Non-communicable Diseases (NCDs), Injuries and Mental Health Program, Africa Centres for Disease Control and Prevention (Africa CDC), Ethiopia; [4]Rwanda Biomedical Centre, University of Global Health Equity (UGHE), Kigali, Rwanda; [5]Division of Disease Control and Prevention, Africa Centres for Disease Control and Prevention (Africa CDC), Ethiopia; [6]Department of Psychology, The Cheikh Anta Diop University (UCAD), Dakar, Senegal; [7]Faculty of Medicine, The Cheikh Anta Diop University (UCAD), Dakar, Senegal; [8]University of Montreal, Montreal, QC, Canada; [9]Institute for Training and Research in Population, Development and Health Reproduction (IPDSR), The Cheikh Anta Diop University (UCAD), Dakar, Senegal; [10]Department of Pediatrics, University of Global Health Equity, Kigali, Rwanda and [11]London School of Hygiene and Tropical Medicine, London, UK

## Abstract

The unavailability of reliable, easy-to-use depression screening tools adapted for Sub-Saharan African children is a significant barrier to the treatment of childhood depression. We thus adapted the Child Depression Screening Tool (CDST) to the South African (SA), Senegalese (S) and Rwandan (R) contexts, as a tool to screen for depression in children suffering from chronic illnesses, trauma and difficulties related to COVID-19, family and community hardships. A DSM-5-based diagnostic interview and the CDST screening measure were administered to 1,001 participants aged between 7 and 16 years. The prevalence of depression ranged between 9.5 and 16.8%. It was more prevalent in youth with chronic illness and those exposed to adverse life events. Older age (R and SA), female sex (S), dislike of school (R and SA) and cannabis use (SA) were also associated with worse depression. Receiver operating characteristic analysis showed satisfactory performance (79-89%) and that sensitivity and specificity were optimized at a CDST cut-point of 5.0. The CDST is a valid tool to screen for depression in the settings assessed. If found to be suitable in other countries and settings, it may offer a clinically sound, sustainable path towards the identification of child depression in Africa.

## Impact statement

- There are few affordable and easy-to-use measures adapted for childhood depression in Africa.
- The Child Depression Screening Tool (CDST), developed in Africa, is a free, rapid screening tool for depression in children that may fill this gap.
- This new tool can contribute to improved identification of depression and referral to appropriate mental health care for children at risk of depression.
- Cannabis use was associated with depression, poor school performance and considerations of dropping out of school. The use of the CDST may provide opportunities to evaluate and treat associated difficulties such as these.

## Background

Mental health difficulties are a major burden for children and adolescents globally, with the World Health Organisation (2021) estimating that ~14% of 10- to 19-year-olds worldwide experience mental disorders. Evidence suggests that depression is one of the most commonly experienced mental disorders in adolescents and that its prevalence is increasing (Daly, 2022; Mojtabai et al., 2016. Around 1.1% of children and adolescents aged 10–14 years, and 2.8% of adolescents aged 15–18 years are estimated to have clinical depression (WHO, 2021a). In Sub-Saharan Africa (SSA), a systematic review encompassing 20 studies reported clinically significant

depressive symptoms in 27% of adolescents in the general population and in 29% of adolescents from at-risk groups (Jorns-Presentati et al., 2021). A more recent review among SSA youth under 19 years of age found a pooled prevalence rate of 15% (Jakobsson et al., 2024). More specifically, in Rwanda, rates of clinically assessed depression in children with human immuno-deficiency virus (HIV) were found to range between 14 and 25% (Binagwaho et al., 2016, 2021). South African (SA) studies have reported that between 4 and 41% of adolescents report experiencing symptoms of depression, potentially indicative of a diagnosis (Pluddemann et al., 2008; Morojele et al., 2013; De Vries et al., 2018).

Risk factors for child and adolescent depression in Africa include biopsychosocial stressors, such as age, sex, food insecurity, bullying and low perceived levels of social support, substance use, poor access to healthcare and exposure to stressful and traumatic events (Partap et al., 2023). Medical risk factors include chronic diseases, such as diabetes, cancer, HIV, tuberculosis and asthma (Harrison et al., 2023). In addition, studies, mostly from high-income countries, report increased levels of depressive symptoms during and after the recent coronavirus disease 2019 (COVID-19) pandemic (Racine et al., 2021; Wang et al., 2022). Youth well-being during this time was likely affected by stress about one's own or loved one's health, social isolation and increased family stressors (*i.e.*, parental job loss and domestic violence) (Loades et al., 2020; Liang and Zeng, 2021; Barendse et al., 2023). Although in SSA youth, low levels of depressive symptoms have been associated with the pandemic, further studies are needed to explore the longer-term effects (Matovu et al., 2021; Wang et al., 2021).

Childhood and adolescent depression are associated with functional impairment in home, school and social domains, as well as increased suicide risk (WHO, 2021b). It is also associated with negative health outcomes in adulthood, such as higher levels of adult anxiety and substance use disorders, worse health and social functioning, less financial and educational achievement and increased criminal behaviour (Johnson et al., 2018; Clayborne et al., 2019; Copeland et al., 2021).

Despite their prevalence and long-lasting effects, child and adolescent mental health and well-being have been overlooked in global health planning (UNICEF, 2021). Most mental health needs in young people are still unmet, especially in low- and middle-income countries (LMICs) where adversity is most prevalent. It is estimated that about four out of five people in LMICs who need services for mental health conditions do not receive them, despite there being effective treatments available (Mangione et al., 2022).

Systematic and scoping reviews have identified several barriers to treatment seeking and accessing professional help for mental health problems. These include limited mental health literacy, perceived social stigma and embarrassment, perceptions around confidentiality and trust of an unknown person, financial costs, resource shortages (*i.e.*, limited access to mental healthcare providers) and logistical barriers (Radez et al., 2021; Saade et al., 2023). Another systematic review, from the primary care providers' perspective, identified barriers related to identification, management and/or referral (O'Brien et al., 2016). A scoping review of barriers specific to African youth found that a preference for traditional or complementary treatments, stigma and mental health literacy was the most common (Saade et al., 2023).

Given the above, there is consensus that child and adolescent mental health services need to be strengthened. This is particularly so in LMICs and SSA, where risk factors may be greater and resources fewer (WHO | Regional Office for Africa, 2021). In a system such as this, it is understandable that many depressed youths may slip through the cracks and not receive the help they need. A brief free screening test will greatly assist in this regard. First, it may assist with the early diagnosis of depression in children and adolescents, allowing them to receive the care that they need to recover. Second, being short and concise, it should not place more of a burden on an already stretched healthcare system where time and capacity are in short supply. Finally, as a free tool, it can be administered without limitations to those children and adolescents who might need it.

In Rwanda, the Children's Depression Inventory (CDI) and the Center for Epidemiological Studies Depression Scale for Children (CES-DC) have been validated with reasonable results (Betancourt et al., 2012; Binagwaho et al., 2016). The CDI, however, requires an administration fee and the CES-DC was not validated in youth with HIV. Thus, Binagwaho et al. undertook to develop a tool that was both free and tailored to young people with HIV (Binagwaho et al., 2021). The Child Depression Screening Tool (CDST) was developed with the support of skilled and knowledgeable local professionals with the assurance that the tool is valid, reliable, affordable and easy for primary care level providers to use (Binagwaho et al., 2021). This approach has the advantage of ensuring that socio-economic and cultural differences are considered, to fully capture the symptoms of depression, ensuring that respondents would fully understand the questionnaire, and that the expression of depression within the Rwandan (R) cultural context is truly actualized (Owen et al., 2016).

Given the positive psychometric results obtained in the R validation, CDST may also offer a clinically sound, sustainable path forward to support the diagnosis and treatment of child depression, particularly in at-risk youth, in SSA. However, for a tool to be used with confidence, validation and adaptation of mental health screening tools for use in a particular setting are crucial to ensure that they accurately identify mental health issues, are culturally appropriate and linguistically accessible (Juhász et al., 2003).

The primary aim of this study was to adapt and validate the CDST, a rapid screening tool, to effectively screen for depression in at-risk children in three SSA countries – Rwanda, Senegal and South Africa (Binagwaho et al., 2021). This included children suffering from HIV and other chronic illnesses, displacement, trauma, as well as experiencing difficulties because of COVID-19, family and community hardships that put them at higher risk of depression (Davidson et al., 2017; Boyes et al., 2019; Awad et al., 2024; Collings and Valjee, 2024). Secondary aims were to assess the prevalence and correlates of depression in the three countries.

## Methodology

### Study design and setting

This was a multi-country cross-sectional study design and was conducted in Rwanda, Senegal and South Africa between December 2021 and March 2022.

### Participants

The sample size calculation was calculated using the Buderer formula and assuming a sensitivity of 88% and a specificity of 96%, based on the results of the initial CDST study conducted in Rwanda (with the cut-off of 6), a 10% width for sensitivity and specificity and a 95% confidence interval (95% CI). The sample size calculation was adjusted for non-response (5–10%), and the sample

size and allocation were adjusted to the study population size in each country to give the following sample sizes: Rwanda $N = 340$, Senegal $N = 500$ and South Africa $N = 300$.

At all sites, we included children aged 7–16 years who gave written assent and whose parents/guardians gave consent to participate. Recruitment took place *via* convenience sampling. Participants from refugee/displacement camps (*e.g.*, youth who left their countries/homes to escape conflict, violence, persecution or natural disaster) were required to have lived in the camp for a minimum of 12 months. No additional inclusion or exclusion criteria were applied. We did, however, select sites where our yield of participants living with chronic diseases (HIV, cancer, diabetes or cardiovascular diseases) and other adverse events (lifetime Diagnostic and Statistical Manual of Mental Disorders, 5th Edition [DSM-5] trauma), recent frightening events (including DSM-5 trauma in the last month) and experiencing COVID or the effects of COVID (*e.g.*, loss of income or close family members) would be high.

In Rwanda, recruitment took place at refugee camps, schools and health facilities. In Senegal, recruitment sites included refugee camps, schools, sites with street-involved youth (*e.g.*, at homeless shelters and with those living and engaged in begging on the streets) and impoverished (poor) youth. In South Africa, recruitment took place at health facilities, children's homes, schools and in communities with high levels of trauma.

### Measures

The data collection tools included a socio-demographic questionnaire, the Child Depression Screening Tool (CDST) (Binagwaho et al., 2021) and a DSM-5-based clinical interview as a gold standard to assess depression. Further, medical data were extracted from patient files, where available. The CDST comprised 11 items, each with 4 response options that are scored from 0 to 3 (0 = *absence of symptoms*, 1 = *symptoms sometimes present*, 2 = *symptoms frequently present* and 3 = *symptoms always present*). The 11 items cover the following areas: mood, representation of the future, interest in games, sleep, fatigue, appetite, attention, agitation, relationships with others and suicidal thoughts. Scores range from 0 to 33, with a cut-point of 6 suggested in the original validation study (Binagwaho et al., 2021).

In Senegal and South Africa, the CDST was translated from English to the local languages by a team of experienced research nurses, clinical psychologists or psychiatrists. To ensure accuracy, the tool was back-translated to English by a different team of clinicians. The translation process had already been completed in Rwanda as part of the development and first validation study (Binagwaho et al., 2021). After testing the tool in small pilot studies and adapting it to each setting, the tool was programmed into the ODK, an open-source Android application, which was used to gather data in electronic format. Data were collected by trained psychologists, nurses and counsellors.

### Procedures

Ethical approvals were obtained before the start of the study.

Children and adolescents who gave written assent and whose parents gave consent to participate were included in the study. The purpose of the study, procedures involved, voluntary nature, potential risks and benefits and assurance of confidentiality of collected information were fully explained, and children were given the option to opt out at the time of the assessment. Measures were made available in the most common languages used in the setting (Rwanda: Kinyarwanda; Senegal: Wolof and French; SA: Afrikaans, English and isiXhosa) to ensure the efficacy and accuracy of the cut-offs obtained.

Psychologists, research nurses and psychological counsellors were trained to administer the CDST and evaluate depression in a standardized manner in the child's preferred language. The interviewer who administered the CDST was blinded to the outcome of the clinical interview and *vice versa*. Children who were identified as requiring further assessment or treatment were referred to mental health clinicians and further specialized services.

### Data analysis

Percentages and 95% CIs were calculated to describe sample characteristics. Scores on the CDST were then compared to MDD diagnoses on the clinical interview to determine sensitivity (proportion of children who have depression according to clinical interview and who are correctly identified by the CDST) and specificity (proportion of children without depression and who have been correctly identified as non-depressed by the CDST) at different cut-points. Receiver operating characteristic (ROC) curve analysis was used to determine the ability of the CDST to discriminate between individuals who did and did not meet the criteria for depression according to the diagnostic interview. The area under the curve (AUC) provides an indication of the diagnostic ability of the CDST: values between 0.5 and 0.7 indicate low discriminatory ability; values between 0.7 and 0.9 indicate moderate discriminatory ability; and values above 0.9 indicate high discriminatory ability of a measure (Hosmer and Lemeshow, 2000).

Assumptions for computing CIs were met as follows: (i) independent observations – visual inspection of our data suggests that each case represents a distinct respondent; (ii) normality – given our sample size, the central limit theorem ensures that the sampling distributions for means, sums and proportions approximate normal distributions.

### Results

### Sample characteristics

A total of 1,001 children and adolescents participated in the study. In Rwanda, 340 children and adolescents participated: (a) 186 (54%) with chronic diseases, (b) 80 (24%) primary and high school children and (c) 75 (22%) children from refugee camps. Their ages ranged between 7 and 15 years. In Senegal, 345 vulnerable youth participated in the study: (a) 151 (43.8%) of these had chronic diseases and (b) 122 (35.4%) were street-involved, refugee and displaced youth, as well as those living in poverty and from schools. Nearly half the children classified as vulnerable were street-involved youth from the capital city of Dakar, approximately one-third were refugee or displaced children and 18% were living in extreme poverty. The most common chronic diseases among the children were sickle-cell anaemia (34%) and HIV (29%), followed by diabetes and cancer. Ages ranged between 7 and 14 years. In South Africa, 315 children and adolescents were included: (a) 9 (2.9%) with a known chronic disease, that is, HIV+, diabetic or direct COVID experience and (b) 84 (26%) who had ever been exposed to DSM-5 trauma and 50 (15.8%) who had experienced a frightening event in the last few weeks. Ages ranged between 7 and 16 years.

## Rwanda

A total of 340 children with a mean age of 11.3 years participated in the study. Table 1. shows that while 88.5% of participants lived with their parents, 11.5% were not living with their parents for various reasons, including the death of parents or separation. The majority of participants (78.8%) were students in primary school and 21.2% were in high school. A considerable proportion of participants had poor academic performance, as 60% repeated a year at least once in their lifetime. In addition, 48.4% missed class time due to health or family reasons, and 9.06% considered dropping out of school.

The mean score on the CDST was 2.9 (95% CI: 1.6, 4.2). Based on the clinical interview, 14.3% (95% CI: 10.9, 18.5) of children were found to have depression. Prevalence was similar in male (14.5%, 95% CI: 10.0, 20.7) and female (14.0%, 95% CI: 9.5, 20.3) participants, but was higher in adolescents aged 13–15 (20.3%, 95% CI: 14.4, 27.9) years than in children in younger age groups (ages 7–9 years = 5.9% 95% CI: 2.5, 13.4 and 10–12 years = 13.3%, 95% CI: 8.1, 20.9). Children living with their parents reported fewer depressive symptoms than those living elsewhere (13.2%, 95% CI: 9.8, 17.6 *vs.* 23.1%, 95% CI: 12.4, 38.8). Higher rates of depression were observed in children not attending school regularly (26.7%, 95% CI: 19.5, 35.3 *vs.* 9.2%, 95% CI: 5.3, 15.5) and in those contemplating dropping out of school (43.5%, 95% CI: 25.1, 63.8 *vs.* 14.9%, 95% CI: 10.8, 20.2).

## Senegal

A sample of 345 participants with a mean age of 11.0 years was included. As shown in Table 1., the vulnerable children included more boys than girls, which largely reflects the much higher number of street-involved boys than girls (52 boys *vs.* 3 girls, respectively). A similar proportion of boys and girls was represented among the students and the children with chronic diseases, except for sickle-cell anaemia (32 boys *vs.* 16 girls).

The overall prevalence of depression determined by a standard clinical interview was 16.8% (95% CI: 13.2, 21.2). Table 1. displays the prevalence of depression for the sample according to socio-demographic characteristics. More girls experienced depression than boys (22.6%, 95% CI: 16.2, 30.5 *vs.* 13.2%, 95% CI: 9.3, 18.5), as did children living in rural versus urban areas (27.8%, 95% CI: 18.6, 39.2 *vs.* 13.9%95% CI: 10.3, 18.6).

The mean CDST score was 4.1 (95% CI: 3.7, 4.4). The prevalence of depression based on clinical interview was particularly high in the Matam and Ziguinchor regions (44.8%, 95% CI: 28.0, 62.9 and 29.3%, 95% CI: 20.4, 40.0, respectively). Children educated in traditional Islamic schools (Daaras) had lower depression prevalence (7.2%, 95% CI: 3.0, 16.3) as compared to those in primary or secondary formal schools (20.3%, 95% CI: 15.1, 26.8), and nearly 16.4% (95% CI: 9.6, 26.8) were boys. Among participants classified as vulnerable children, children with chronic diseases and students, depression prevalence was 16.1% (95% CI: 10.5, 23.9), 19.2% (95% CI: 13.7, 26.3) and 13.2% (95% CI: 7.2, 22.8), respectively. While sample sizes are small when stratified by type of vulnerability, results suggest that refugee (29.2%, 95% CI: 11.0, 47.4) and displaced (27.3%, 95% CI: 8.7, 45.8) children are more likely to suffer from depression compared with street-involved youth (9.1%, 95% CI: 1.5, 16.7) and children living in extreme poverty (9.5%, 95% CI: −3.0, 22.1). More than one in five children with cancer, sickle-cell anaemia and HIV were identified to have depression. When we stratified results by sex, vulnerable girls had a particularly elevated prevalence of depression (32.3%, 95% CI: 15.8, 48.7) compared to boys (10.3%, 95% CI: 3.9, 16.7).

## South Africa

In South Africa, 315 participants with a mean age of 11.6 years were included. Just over a quarter, 26.7%, of the 315 children endorsed lifetime trauma exposure, and 15.9% had experienced a frightening event in the last few weeks. Only a few children had direct exposure to COVID-related trauma, with 1.0% indicating that they had lost a close family member to COVID. Of the 2.9% who indicated that they were aware of having a chronic disease, 1.9% indicated that they were HIV+ and 1.0% indicated that they were diabetic. A large number of participants were from disadvantaged environments, with just over a third (36.5%) indicating that their family received a government grant (*e.g.*, disability grant or pension). The majority (96.1%) resided in urban areas in the Cape Metropole region of South Africa. While 27.8% had lived away from home for more than 3 months at some time, at the time of this study, only 23.2% were currently living in a group or boarding home. Socio-demographic characteristics of participants are displayed in Table 1.

The mean CDST score was 5.4 (95% CI: 4.9, 5.9). The number of children who scored above the recommended cut-off of 6 on the CDST (Binagwaho et al., 2021), that is, those with probable depression, was 22 (26.7%). The prevalence of depression, as determined by clinicians conducting the clinical interviews, was 9.5%. A diagnosis of major depressive disorder was more prevalent in older children (13–16 years old [16.8%, 95% CI: 11.6, 23.7]), than in younger children (7–9 years old [1.5%, 95% CI: 0.2, 9.9] and 10–12 years old [3.1%, 95% CI: 1.0, 9.1]). Children who indicated that they considered dropping out of school (27.8%, 95% CI: 22.8, 32.8 *vs.* 8.1%, 95% CI: 5.1, 11.1) and children who had lost their mothers were more likely to be depressed (20.7% 95% CI: 9.6, 39.2 *vs.* 8.4% 95% CI: 5.7, 12.3). Those who experienced a recent frightening event (22%, 95% CI: 17.4, 26.6 *vs.* 95% CI: 69.9, 86.1) and those who had experienced COVID or had a chronic illness (33.3%, 95% CI: 28.1, 38.5 *vs.* 57.8, 76.2) were also more likely to be depressed. Of note, almost half (43%) of the participants who smoked cannabis were depressed. Cannabis use was related to age, with older children more likely to be using the substance (95% CI: −0.003, 0.052).

## Criterion validity of the CDST

CDST scores were compared to the clinical interview results to obtain sensitivity and specificity. Table 2 provides the sensitivity and specificity of different scores per country. For all three countries, Rwanda, Senegal and South Africa, the cut-point of 5 provided the best sensitivity and specificity.

The AUC is used to assess the overall performance of a test. In ROC analyses, the CDST showed good discriminatory power relative to the DSM-5-based structured clinical interview for depression, with an AUC of 0.90 for Rwanda, 0.89 for Senegal and 0.79 for South Africa. See Figures 1–3. These AUCs of above 0.79 indicate that the CDST performed significantly better than chance at discriminating between those with and without depression in the three countries.

**Table 1.** Socio-demographic characteristics of the samples: Rwanda, South Africa and Senegal

| | Rwanda | | | | Senegal | | | | South Africa | | | |
|---|---|---|---|---|---|---|---|---|---|---|---|---|
| | n | Percent | Prevalence of depression | 95% Cl | n | Percent | Prevalence of depression | 95% Cl | n | Percent | Prevalence of depression | 95% Cl |
| **Overall** | **336** | | **14.3** | **[10.9,18.5]** | **345** | | **16.8** | **[13.2,21.2]** | **315** | | **9.2** | **[6.5,13.0]** |
| **Age group (years)** | | | | | | | | | | | | |
| 7–9 | 85 | 25.3 | 5.9 | [2.5,13.4] | 110 | 31.90 | 16.4 | [10.5,24.5] | 67 | 21.3 | 1.5 | [0.2,9.9] |
| 10–12 | 113 | 33.6 | 13.3 | [8.1,20.9] | 130 | 37.70 | 16.2 | [10.8,23.5] | 98 | 31.2 | 3.1 | [1.0,9.1] |
| 13–14/15/16 | 138 | 41.1 | 20.3 | [14.4,27.9] | 105 | 30.40 | 18.1 | [11.8,26.7] | 149 | 47.5 | 16.8 | [11.6,23.7] |
| **Sex** | | | | | | | | | | | | |
| Male | 172 | 51.2 | 14.5 | [10.0,20.7] | 212 | 61.40 | 13.2 | [9.3,18.5] | 151 | 3.8 | 7.9 | [4.6,13.5] |
| Female | 164 | 48.8 | 14 | [9.5,20.3] | 133 | 38.60 | 27.8 | [16.2,30.5] | 164 | 96.2 | 11.0 | [7.0,16.8] |
| **Parents are alive** | | | | | | | | | | | | |
| Both parents alive | | | | | 296 | 85.90 | 15.9 | [12.1,20.5] | | | | |
| At least one parent deceased | | | | | 49 | 14.10 | 22.4 | [12.9,36.2] | | | | |
| **Is your mom alive?** | | | | | | | | | | | | |
| Yes | | | | | | | | | 285 | 90.8 | 8.4 | [5.7,12.3] |
| no | | | | | | | | | 29 | 9.2 | 20.7 | [9.6,39.2] |
| **Is your dad alive?** | | | | | | | | | | | | |
| Yes | | | | | | | | | 247 | 80.2 | 8.5 | [5.6,12.7] |
| no | | | | | | | | | 61 | 19.8 | 14.8 | [7.8,26.1] |
| **What is your parents' marital situation?** | | | | | | | | | | | | |
| Married | | | | | | | | | 57 | 19.4 | 14.0 | [7.1,25.7] |
| Divorced | | | | | | | | | 12 | 4.1 | 16.7 | [4.2,47.9 |
| Never married/single | | | | | | | | | 201 | 68.4 | 7.0 | [4.2,11.4] |
| Widow(ed) | | | | | | | | | 24 | 8.2 | 8.3 | [2.1,28.1] |
| **Child live with both parents** | | | | | | | | | | | | |
| Yes | 296 | 88.1 | 13.2 | [9.8,17.6] | | | | | | | | |
| No | 39 | 11.6 | 23.1 | [12.4,38.8] | | | | | | | | |
| **Place of residence** | | | | | | | | | | | | |
| Urban | | | | | 273 | 79.10 | 13.9 | [10.3,18.6] | 306 | 96.1 | 9.8 | [6.5,13.1] |

(*Continued*)

**Table 1.** (*Continued*)

| | Rwanda | | | | Senegal | | | | South Africa | | | |
|---|---|---|---|---|---|---|---|---|---|---|---|---|
| | *n* | Percent | Prevalence of depression | 95% Cl | *n* | Percent | Prevalence of depression | 95% Cl | *n* | Percent | Prevalence of depression | 95% Cl |
| Rural | | | | | 72 | 20.90 | 27.8 | [18.6,39.2] | 9 | 3.9 | 0 | [0] |
| **Region** | | | | | | | | | | | | |
| Dakar | | | | | 171 | 49.60 | 9.4 | [5.8,14.8] | | | | |
| Matam | | | | | 29 | 8.40 | 44.8 | [28.0,62.9] | | | | |
| Saint Louis | | | | | 63 | 18.30 | 7.9 | [3.3,17.8] | | | | |
| Ziguinchor | | | | | 82 | 23.80 | 29.3 | [20.4,40.0] | | | | |
| **Number of meals per day** | | | | | | | | | | | | |
| One | | | | | 16 | 4.60 | 25.0 | [9.7,51.0] | | | | |
| Two | | | | | 39 | 11.30 | 17.9 | [8.8,33.2] | | | | |
| Three or more | | | | | 290 | 84.10 | 16.2 | [12.4,20.9] | | | | |
| **Religion of the child** | | | | | | | | | | | | |
| Christian | | | | | | | | | 256 | 82.3 | 10.2 | [7.0,14.5] |
| Muslim | | | | | | | | | 21 | 6.8 | 4.8 | [0.7,27.4] |
| None/others | | | | | | | | | 28 | 10.9 | 7.1 | [1.8,24.6] |
| **Child live in a boarding school** | | | | | | | | | | | | |
| No | 255 | 76.8 | 17.3 | [13.1,22.4] | | | | | 241 | 76.8 | 9.1 | [5.9,12.3] |
| Yes | 71 | 21.1 | 2.8 | [0.7,10.6] | | | | | 73 | 23.2 | 11.0 | [7.5,14.5] |
| **Current education level** | | | | | | | | | | | | |
| Primary | 266 | 79.1 | 14.3 | [10.6,19.1] | 182 | 52.80 | 20.3 | [15.1,26.8] | 215 | 68.7 | 4.7 | [2.5,8.5] |
| Secondary | 70 | 20.8 | 14.3 | [7.8,24.6] | 73 | 21.20 | 16.4 | [9.6,26.8] | 98 | 31.3 | 20.4 | [13.5,29.6] |
| Daara (Islamic school) | | | | | 69 | 20.00 | 7.2 | [3.0,16.3] | | | | |
| No formal education | | | | | 21 | 6.10 | 19.0 | [7.3,41.3] | | | | |
| **Ever repeated school year** | | | | | | | | | | | | |
| No | 151 | 44.9 | 18.5 | [13.1,25.6] | | | | | | | | |
| Yes | 101 | 30.1 | 15.8 | [9.9,24.4] | | | | | | | | |

(*Continued*)

**Table 1.** (*Continued*)

| | Rwanda | | | | Senegal | | | | South Africa | | | |
|---|---|---|---|---|---|---|---|---|---|---|---|---|
| | *n* | Percent | Prevalence of depression | 95% Cl | *n* | Percent | Prevalence of depression | 95% Cl | *n* | Percent | Prevalence of depression | 95% Cl |
| **Are you often absent from school?** | | | | | | | | | | | | |
| Yes | 120 | 35.7 | 26.7 | [19.5,35.3] | | | | | 36 | 27.8 | 8.3 | [2.7,23.0] |
| No | 131 | 39.0 | 9.2 | [5.3,15.5] | | | | | 278 | 72.2 | 9.7 | [6.7,13.8] |
| **Considered dropping out of school** | | | | | | | | | | | | |
| Yes | 23 | 6.8 | 43.5 | [25.1,63.8] | | | | | 18 | 5.8 | 27.8 | [22.8,32.8] |
| No | 228 | 67.9 | 14.9 | [10.8,20.2] | | | | | 295 | 94.2 | 8.1 | [5.1,11.1] |
| **Population type** | | | | | | | | | | | | |
| Students | | | | | 76 | 22.0 | 13.2 | [7.2,22.8] | | | | |
| Vulnerable children (street-involved, refugee/displaced, impoverished | | | | | 118 | 34.20 | 16.1 | [10.5,23.9] | | | | |
| Street-involved | | | | | 55 | 45.10 | 9.1 | [1.5,16.7] | | | | |
| Refugee | | | | | 24 | 19.70 | 29.2 | [11.0,47.4] | | | | |
| Displaced | | | | | 22 | 18.00 | 27.3 | [8.7,45.8] | | | | |
| Impoverished | | | | | 21 | 17.20 | 9.5 | [−3.0,22.1] | | | | |
| Children with chronic disease | | | | | 151 | 43.80 | 19.2 | [13.7,26.3] | | | | |
| Ever exposed to DSM–5 trauma | | | | | | | | | 84 | 26.7 | 11.9 | [5.6,18.4] [81.6, 94.4] |
| Recent frightening event | | | | | | | | | 50 | 15.8 | 22 | [17.4,26.6] 69.9,86.1 |
| HIV+, diabetic + or had COVID | | | | | | | | | 9 | 2.9 | 33.3 | [28.1,38.5] 57.8, 76.2 |

**Table 2.** Sensitivity and specificity of the CDST at different cut-points for Rwanda, Senegal and South Africa

|  | Sensitivity | Specificity |
|---|---|---|
| **Rwanda** | | |
| Score – 4 | 81% | 90% |
| Score – 5 | 81% | 95% |
| Score – 6 | 75% | 97% |
| Score – 7 | 65% | 99% |
| Score – 8 | 63% | 99% |
| **Senegal** | | |
| Score – 4 | *91%* | *61%* |
| Score – 5 | 90% | 75% |
| Score – 6 | 74% | 84% |
| Score – 7 | 62% | 90% |
| Score – 8 | 53% | 93% |
| Score – 9 | 41% | 96% |
| **South Africa** | | |
| Score – 4 | 80% | 59% |
| Score – 5 | 80% | 71% |
| Score – 6 | 70% | 78% |
| Score – 7 | 53% | 82% |
| Score – 8 | 47% | 85% |

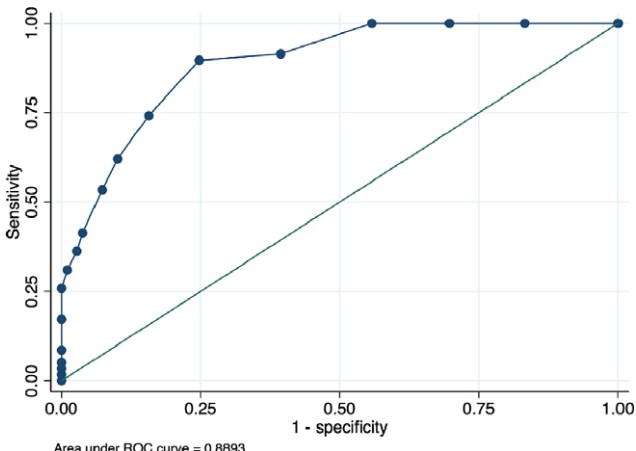

**Figure 2.** Receiver operating characteristic curve of the CDST for Senegal.

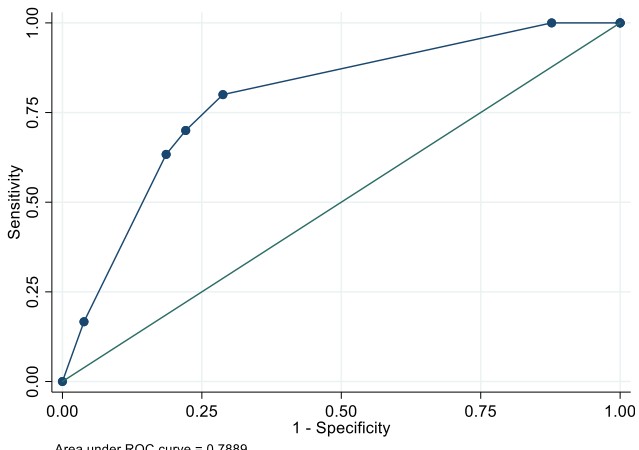

**Figure 3.** Receiver operating characteristic curve of the CDST for South Africa.

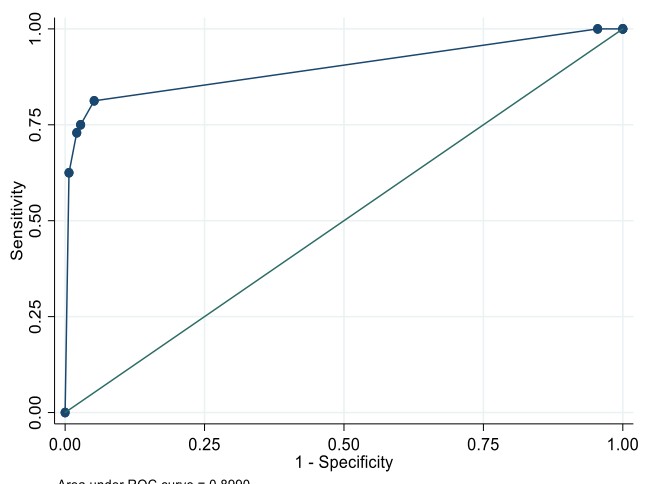

**Figure 1.** Receiver operating characteristic curve of the CDST for Rwanda.

## Discussion

Although numerous tools to screen for depression in children are available, few are accessible and adapted to African settings. This cross-sectional study aimed to adapt the CDST, a rapid tool to screen for childhood depression. Accurate assessments can be critical to targeting resources, especially when there are limited resources for mental health treatment. A total sample of 1,001 children and adolescents was recruited across three countries (Rwanda: *n* = 340; Senegal: *n* = 345; South Africa: *n* = 316).

ROC analysis was conducted to identify the CDST cut-point that best predicted depressive status as assessed by the clinical interview. The ROC curve demonstrated that sensitivity and specificity in all three samples were optimized at a cut-point of 5.0. This is one point lower than the recommended cut-off of 6, based on the original validation study (Binagwaho et al., 2021). At the cut-point of 5, sensitivity was highest in the Senegalese (S) sample (90%) as compared to the R (81%) and SA (80%) samples. Specificity at this cut-point was best in the R sample (95%) as compared to the S (75%) and SA (71%) samples. In addition, the performance of the measure, according to the ROC analysis, was satisfactory, which was 89–90% (medium–high) in the R and S samples and 79% (medium) in the SA sample. These robust AUC statistics indicate that depressed youth are 79–90% more likely to have a high total score on the CDST than those who do not have depression. The sound psychometric properties and anecdotal ease of use expressed by those who administered the CDST suggest that the CDST can be a useful tool to screen for depression in children and adolescents in these settings.

Secondary aims included estimating the prevalence and correlates of depression in the three countries. Based on diagnostic clinical interviews, the prevalence of depression was lowest in SA youth (9.3%) as compared to R and S youth (14.3% and 16.8%,

respectively). Given the heterogeneity of the youth samples in the three countries, these prevalence estimates cannot be directly compared. The sample from South Africa was predominantly composed of children exposed to family and community hardships and had a lower number of children diagnosed with chronic illnesses — <3% of the sample as compared to 54% and 43.8% in the R and S samples, respectively.

In accordance with studies showing an association between chronic disease and depression (Binagwaho et al., 2021; Dessauvagie et al., 2020; Too et al., 2021), elevated rates of depression were found in youth with chronic illness in all three countries. The risk of depression was also significantly higher among children and adolescents exposed to adverse life events, such as the death of a family member, physical or sexual abuse or being a refugee, in all countries. Exposure to adverse events such as these has consistently been identified as a risk factor for depression (Rao and Chen, 2009; Thapar et al., 2012; Oldehinkel et al., 2015; Beck et al., 2021; Jorns-Presentati et al., 2021).

The prevalence rates are similar to global prevalence rates and a study done in Ethiopia (Belfer, 2008; Girma et al., 2021; Racine et al., 2021). Although lower than those found in other African countries (*e.g.*, Uganda [21%] and Nigeria [21.2%]), these studies based their findings on self-report measures, which are known to provide higher estimates (Fatiregun and Kumapayi, 2014; Nalugya-Sserunjogi et al., 2016). We found that older children/adolescents were more likely to be at risk of depression than were younger children. Numerous studies have confirmed this finding of adolescents being more at risk than children, possibly due to the emotional, psychological and physical changes that they undergo during this developmental period (Belfer, 2008; Costello et al., 2011; Jorns-Presentati et al., 2021; Oldehinkel et al., 2015; Racine et al., 2021).

Girls had a higher rate of depression than boys in the S sample. Female sex has commonly been found to be a risk factor for depression, including in LMICs, such as Ethiopia, India and Uganda (Patten et al., 2006; Nalugya-Sserunjogi et al., 2016; Trivedi et al., 2016; Riecher-Rössler, 2017; Beck et al., 2021; Girma et al., 2021; Racine et al., 2021; Too et al., 2021). However, this was not so in both the SA and R samples. This may be explained by the younger age of participants, as while similar rates of depression have been found during childhood, females are at increased risk during and after adolescence (Hyde et al., 2008; Alsaad et al., 2022).

Depression has been associated with a number of long-term psychosocial outcomes. These include a lower likelihood of entering post-secondary education, poor performance at school, an increased risk of leaving secondary school and substance abuse (Dunn and Goodyer, 2006; Lund et al., 2010; Cairns et al., 2014; Maras et al., 2015; Gunnell et al., 2016; Clayborne et al., 2019; Beck et al., 2021; Olisaeloka et al., 2024; Ward-Smith et al., 2024). Similarly, we found that those who indicated that they disliked school and those who considered dropping out of school were more likely to be depressed, and, in the SA sample, close to half of those who smoked cannabis were depressed.

Recent reviews and meta-analyses of cannabis use in SSA adolescents have reported rates of 4–8% (Asante and Atorkey, 2023; Belete et al., 2023). A 2007 review of cannabis use in South Africa reported a current self-reported rate of 5–10% among adolescents (Peltzer and Ramlagan, 2007). Although the rate of cannabis use in this sample was lower, almost half of those who did use met criteria for depression. Systematic reviews and a meta-analysis determined that cannabis use in adolescence is associated with both higher levels of depression and predictive of depression, with some reporting that the links between heavy cannabis use during adolescence and poorer academic success and educational attainment are thought to be associated with lower academic motivation (Cairns et al., 2014; Pacheco-Colón et al., 2019).

The findings should be viewed in light of the study's limitations. First, as the samples from each country were largely convenience-based, they cannot be considered representative of the populations they were drawn from. Second, we unfortunately did not capture data on the frequency and length of substance use; this would be important to include in future studies. Third, we unintentionally omitted capturing the language in which the CDST was administered; this could have provided useful information regarding cut-off scores in each of the languages. Despite this, strengths of this study include that the validation process and assessment of predictors remain substantially robust, and that the CDST is developed in Africa and is a free, open-access rapid assessment tool for depression. In addition, anecdotally, the researchers who administered the CDST found it easy to use in all three countries. Thus, the CDST can allow early diagnosis as a first step towards access to treatment for depression management in Rwanda, Senegal and South Africa.

## Conclusion

This study demonstrates the validity of the CDST in Rwanda, Senegal and South Africa. If found to be valid and reliable in other African settings, it may be used to enhance the capacity of community-based healthcare providers to identify and refer youth with depression. In addition, given the association between cannabis use and depression, as well as cannabis use and poorer school performance/considerations of dropping out of school, the use of the CDST and similar tools may open up possibilities for healthcare professionals and community health workers to evaluate and treat these associated difficulties and conditions. Accurate and early identification of symptoms that take socio-economic and cultural differences into account can facilitate referral for appropriate treatment and improve long-term well-being (Patton et al., 2016).

**Open peer review.** To view the open peer review materials for this article, please visit http://doi.org/10.1017/gmh.2025.10022.

**Data availability statement.** The data that support the findings of this study are available from the corresponding author, SS, upon reasonable request.

**Acknowledgements.** Investigation: Maryke Hewett, Bulelwa Zibi, Irene Mbanga, Letticia Hintsho, Alexandra Kreuz Goldberg, Lynn Aupais, Dedri Hamman, Erin Jill Hector, Siphokazi Mnconywa, Emily van der Westehuizen and Anusha Lachman.
Methodology: Maryke Hewett. Formal analysis: Martin Kidd.

**Author contribution.** Sharain Suliman: Formal analysis, investigation, methodology, project administration, supervision, writing – original draft, writing – review and editing.
Jenny Bloom: Methodology, project administration, supervision, writing – review and editing.
Naeem Dalal: Writing – review and editing.
Eric Remera: Data curation, formal analysis, project administration, supervision, writing – review and editing.
Raissa Muvunyi: Data curation, formal analysis, project administration, supervision, writing – review and editing.
Mohammed Abdulaziz: Funding acquisition, project administration, writing – review and editing.

Adelard Kakunze: Funding acquisition, project administration, writing – review and editing.
Ismahan Soukeyna Diop: Investigation, writing – review and editing.
Djena Fafa Cisse: Investigation, writing – review and editing.
Awa Der Dieye: Investigation, writing – review and editing.
Britt McKinnon: Formal analysis, methodology, supervision, writing – review and editing.
Mohamadou Sall: Conceptualization, funding acquisition, methodology, project administration, resources, supervision, writing – review and editing.
Agnes Binagwaho: Conceptualization, funding acquisition, methodology, project administration, resources, supervision, writing – review and editing.
Soraya Seedat: Conceptualization, funding acquisition, methodology, project administration, resources, supervision, writing – review and editing.

**Financial support.** This work was supported by the Non-communicable Diseases (NCDs), Injuries and Mental Health Program, Africa Centres for Disease Control and Prevention (Africa CDC).

**Competing interest.** The authors declare none.

**Ethics statement.** Ethics approvals were obtained before the start of the research: Rwanda – Rwanda National Ethics Committee (ref: No.406/RNEC/2019) and the Institutional Review Board of University of Global Health Equity; Senegal – Cheikh Anta Diop University (ref: 00000209/MSAS/CNERS/SP); South Africa – Stellenbosch University Faculty of Medicine and Health Sciences (HREC ref.: N21/10/111).

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
