## [Reviewer Report]

Excellent paper on the development of a mental health screening instrument for use in LMIC of Sub-Saharan Africa. Comparison with clinical interviews suggest that the CDST is a valid instrument. Well done!

---

## [Reviewer Report]

Validation of the Child Depression Screening Tool in Three African Settings: Rwanda, Senegal and South Africa – Peer review comments

Thank you for the opportunity to review this paper. This manuscript provides valuable insights into the validation of the Childhood Depression Screening Tool (CDST) across three Sub-Saharan African settings. It addresses an important gap in the availability of contextually adapted, accessible mental health screening tools for vulnerable children in low-resource contexts. However, several areas require improvement to enhance the clarity, consistency, and impact of the paper. Key issues include providing more comprehensive rationales and justifications in the introduction, methodological clarifications, consistency in the presentation of results, and ensuring statistical accuracy. The suggested revisions aim to improve the manuscript while maintaining its important findings.

Major Comments (by section)

Introduction

1. Including up-to-date prevalence statistics in the three settings would provide additional context and comparisons for current results. For example, Rwanda reports rates ranging from 14.2% (https://doi.org/10.1186/s12887-020-02475-1) to 25% (https://doi.org/10.1186/s12887-016-0565-2). Including a more up-to-date systematic review and meta-analysis (https://doi.org/10.1017/gmh.2024.82) would also be helpful.

2. The section on risk factors (page 5 lines 16-23) would be improved by including a comprehensive review of psychosocial stressors, such as food insecurity (e.g., https://doi.org/10.1111/tmi.13336), bullying victimisation (https://doi.org/10.1017/S2045796022000683; https://doi.org/10.1016/j.pmedr.2023.102499), exposure to violence (https://doi.org/10.1111/tmi.13336), etc. Left as is, the section might seem to “cherry-pick” the psychosocial stressors the paper subsequently includes as sociodemographic characteristics.

3. The section on risk factors would also be clearer if it distinguished between psychosocial and medical risk factors more clearly, as it currently includes chronic diseases as psychosocial stressors.

4. The section on COVID-19-related prevalence and factors (lines 19–23) would benefit from citing studies conducted in sub-Saharan Africa, such as https://doi.org/10.4269/ajtmh.20-1620).

5. The paper would be more aligned with its scope if the discussion of barriers to treatment seeking focused on studies carried out in or about sub-Saharan Africa rather than Europe/North America e.g., https://doi.org/10.1186/s12913-023-09294-x.

6. The statement that “the majority of tools presently available are either insufficient or not affordable in the SSA context” (page 7 lines 10-11) would be strengthened by providing references and discussing alternatives like the CES-DC (https://doi.org/10.1016/j.jaac.2012.09.003) and CES-D 10 (https://doi.org/10.1186/s12887-016-0565-2). Including a rationale for choosing to validate this instrument over others would further clarify this point.

7. The introduction to the CDST (page 7 lines 11-19) would be improved by contextualising claims about its advantages, particularly noting that its validity and reliability have only been demonstrated for children affected by HIV in Rwanda so far.

8. Including a rationale for validating screening tools before use would justify the paper’s aims and objectives. This rationale could then reinforce the claims made about the tool ensuring that cultural differences are considered to fully capture the symptoms. A rationale for using the CDST with children affected by displacement, transgenerational trauma, and other hardships could also justify the paper’s aims further.

Methods

1. The methods section should describe the recruitment strategy. Since the discussion mentions convenience-based sampling, this should be stated explicitly.

2. The Participants section could be rewritten for clarity and would benefit from differentiating recruitment sites across the three countries. The section should also include inclusion and exclusion criteria.

3. Providing a detailed description of the CDST, such as the number of items and scoring methods, the cut-off point in the original validation study… would enhance understanding of the tool and its use.

• The sample size calculation should be included to clarify the study’s statistical robustness. Ensuring stratified sample sizes are sufficient for conclusions would increase confidence in the findings.

4. The methods section should also state what type of clinical interview was utilised. The results section mentions the MINI Kid clinical interview was used and this should be stated explicitly here. The choice of the MINI Kid clinical interview should also be justified by explaining why it was preferred over alternatives.

5. The discussion states that practitioners found the CDST easy to use in all three countries. The methods to reach these results should be included in this section.

6. Defining “vulnerable children” more clearly and explaining how the population subtypes (e.g., street-involved, refugee, impoverished, recent frightening event, direct COVID experience) were operationalised would improve clarity and contextualisation of results.

7. Generally, justification for the methods chosen would strengthen the methods section.

8. Importantly, the results and discussion sections mention significant differences in prevalence by different sociodemographic characteristics. What statistical test was used to reach these conclusions? This should be stated in this section more clearly. As it is, it mentions prevalence rates, ANOVAs, chi-square tests and correlations but does not specify what test was used to generate what results. Assumptions testing should also be discussed here.

9. Page 10 lines 16-19: a reference for this interpretation should be included.

Results

1. Clarifying the sociodemographic categories and how they are operationalised (e.g., differentiating “street-involved, refugee, displaced, impoverished, extreme poverty” categories) would improve understanding of these.

2. The classification of “direct COVID experience” as a chronic disease could be reconsidered for accuracy.

3. Including percentages, confidence intervals, and p-values for comparisons (e.g., age group differences in depression rates) would strengthen the results’ validity. It could be problematic to state “significant differences” (e.g., “Prevalence was… significantly higher in adolescents aged 13-15 years than in children in younger age groups”) without including correct statistics and significance levels.

4. Some statistics are reported approximately or not at all (e.g., “Prevalence was approximately similar in male and female participants, but was significantly higher in adolescents aged 13-15 years than in children in younger age groups”). Consider including specific percentages, p-values and 95% CIs throughout.

5. Consistency in reporting across countries (e.g., sample sizes, mean CDST scores) would enhance coherence. E.g., the mean CDST scores are just mentioned for South Africa while it would be interesting to include it for the other two settings as well.

6. The AUC statistic for Rwanda according to Figure 1 is 0.90 instead of 0.89 which is stated throughout the paper. Please ensure the correct figure is stated throughout.

Discussion

1. Statements about the benefits of using the CDST (e.g., “can allow for early diagnosis, as well as better access to treatment and follow-up for depression management”) should be justified and backed up by references.

2. The discussion would be strengthened by comparing findings to prior research e.g., current prevalences to previously reported country-specific prevalences (instead of global statistics or statistics from other African countries); current performance characteristics to the ones in the original validation study…)

Conclusion

1. Similarly to the discussion section, statements about the benefits of using the CDST (e.g., “to enhance the capacity of community-based health-care providers to identify, diagnose, and effectively treat youth with depression…. will strengthen the mental health referral processes”…), should be justified and backed up by references.

Table 1

1. I recommend a thorough revision of Table 1 (and therefore of the results and results section) as some calculations seem to be wrong (e.g., most values in the percentages columns are over 100 in the results on Rwanda), some seem to be missing (e.g., the prevalence of depression in Rwanda if they were having problems at school), there is a p-value in the 95%CI column…

2. Are the asterisks in the table meant to be there? If so, it would be helpful to include a description of what they mean.

Minor Comments

1. Page 2 Lines 21-24: This section could be rewritten for clarity.

2. Page 4 line 15: Consider stating the performance statistics are related to the AUC.

3. Page 7 lines 10-11: Including a reference for the following statement: “The majority of tools presently available are either insufficient or not affordable in the SSA context” would increase confidence in its veracity.

4. Page 11 lines 2, 8-9, and 12: Consider including the mean age of participants to aid understanding.

5. I recommend being consistent when reporting e.g., in page 12 line 11 you say “(95% confidence interval (CI): 13.2, 21.2)”, in Page 11 line 22 you say “(95% CI: 10.9, 18.5)”, in page 13 line 13 you say “5.4 (±4.6)” …

6. Page 14 line 5: consider changing the subtitle “Validation of the CDST” to something more appropriate e.g., “Criterion validity of the CDST”.

7. Page 14 line 16: consider including a reference for these interpretations.

8. Page 17 lines 5-6: consider stating the percentages refer to AUC values.

9. Table 1: Keeping decimal places consistent, as some figures are to no decimal places, some to one and some to two, would make the table look more coherent.

Overall, these suggestions aim to support the authors in presenting their important findings with greater accuracy, clarity and impact. Thank you again for the opportunity to review this work.

---

## [Editor Report]

Dear Authors,

Your manuscript: “Validation of the Child Depression Screening Tool in Three African Settings: Rwanda, Senegal and South Africa”, has now been reviewed,

---

## [Reviewer Report]

Thank you for your thoughtful responses to my comments and for the revisions made to the manuscript. I’m satisfied with the changes and believe they have strengthened the article. I have no further suggestions.

---

## [Editor Report]

Dear Suliman Sharain,

Your revised manuscript ‘Validation of the Child Depression Screening Tool in Three African Settings: Rwanda, Senegal and South Africa’ has been reviewed